# PharmaKU: A Web-Based Tool Aimed at Improving Outreach and Clinical Utility of Pharmacogenomics

**DOI:** 10.3390/jpm11030210

**Published:** 2021-03-16

**Authors:** Sumi Elsa John, Arshad Mohamed Channanath, Prashantha Hebbar, Rasheeba Nizam, Thangavel Alphonse Thanaraj, Fahd Al-Mulla

**Affiliations:** Genetics & Bioinformatics, Dasman Diabetes Institute, 15462 Dasman, Kuwait; sumi.john@dasmaninstitute.org (S.E.J.); arshad.channanath@dasmaninstitute.org (A.M.C.); prashantha.hebbar@dasmaninstitute.org (P.H.); rasheeba.iqbal@dasmaninstitute.org (R.N.); alphonse.thangavel@dasmaninstitute.org (T.A.T.)

**Keywords:** pharmacogenomics, precision medicine, star alleles, dosage recommendation, bioinformatics, whole genome sequencing, next-generation sequencing, genomics

## Abstract

With the tremendous advancements in genome sequencing technology in the field of pharmacogenomics, data have to be made accessible to be more efficiently utilized by broader clinical disciplines. Physicians who require the drug–genome interactome information, have been challenged by the complicated pharmacogenomic star-based classification system. We present here an end-to-end web-based pharmacogenomics tool, PharmaKU, which has a comprehensive easy-to-use interface. PharmaKU can help to overcome several hurdles posed by previous pharmacogenomics tools, including input in hg38 format only, while hg19/GRCh37 is now the most popular reference genome assembly among clinicians and geneticists, as well as the lack of clinical recommendations and other pertinent dosage-related information. This tool extracts genetic variants from nine well-annotated pharmacogenes (for which diplotype to phenotype information is available) from whole genome variant files and uses Stargazer software to assign diplotypes and apply prescribing recommendations from pharmacogenomic resources. The tool is wrapped with a user-friendly web interface, which allows for choosing hg19 or hg38 as the reference genome version and reports results as a comprehensive PDF document. PharmaKU is anticipated to enable bench to bedside implementation of pharmacogenomics knowledge by bringing precision medicine closer to a clinical reality.

## 1. Introduction

Pharmacogenomics (PGx) is the field that studies how genetic makeup affects a person’s response to drugs. Even though the concept of pharmacogenomics has been around since the 1950s [1], it is only now that we witness its proper integration with clinical informatics for clinical decision support (CDS). Advancements in array-based and high-throughput sequencing technologies have enabled scientists to quickly profile an individual’s genetic make-up, which can be used to query pharmacogenomics resources. Pharmacogenomics is a good example of integration of Precision Medicine in medical practice [2]. By way of profiling an individual’s genetic make-up through array-based and high throughput sequencing technologies, it is now possible to predict if a specific medicine will be effective in a person or likely to cause adverse drug reactions (ADRs).

Pharmacogenomics plays an instrumental role in drug safety and efficacy. Studies indicate that the most commonly prescribed pharmaceuticals are effective in only 25% to 60% of patients [3]. Furthermore, each year, hospitals in the United States report more than two million patients with ADRs, resulting in up to 100,000 fatalities and a total cost of up to $5.6 million per hospital [4]. In a multicenter study by Pirmohamed et al., ADRs were found to account for 6.5% of hospitalizations in two large hospitals in the United Kingdom [5]. Interestingly, almost 100% of the population carries at least one actionable genetic variant [6,7]. Haplotypes are groups of variants in a person’s genome that are inherited together. Some of the conditions known to affect a person’s response to certain drugs include warfarin resistance [8], warfarin sensitivity [9], clopidogrel resistance [10], malignant hyperthermia [11], Stevens–Johnson syndrome/toxic epidermal necrolysis [12] and thiopurine S-methyltransferase deficiency [13].

Genetic variants together with environmental factors play an important role in an individual’s response to drug treatment. As sequencing has become more affordable, many health centers can now easily get patient genomes sequenced. Genetic markers in pharmacogenomics are identified by means of numbers and letters and separated from gene names by a star known as star allele nomenclature. For example, CYP2B6*2 identifies the genetic variant in gene CYP2B6 at genomic position g.5071C > T, leading to amino acid substitution R22C [14]. Star allele nomenclature has become the gold standard in pharmacogenomics as it helps standardize the identification of pharmacogenetic alleles better and helps to avoid transcription mistakes, which are more common when using Human Genome Variation Society (HGVS) nomenclature. In pharmacogenomics, accurate detection of star alleles in clinically actionable pharmacogenes provides the foundation for phenotype prediction and treatment decisions.

Custom-designed pharmacogenomic arrays were the technology of choice for their ability to provide faster, cost-effective solutions, particularly for large sample sizes as part of research studies. The Affymetrix-developed Drug Metabolizing Enzymes and Transporters (DMET) Plus array [15] was one of the first pharmacogenomic arrays, which was implemented in two PGx initiatives, the 1200 patients Project [16] and the PG4KDS protocol [17]. Microarrays were successfully deployed in several other pre-emptive pharmacogenomics initiatives such as Pharmacogenomic Resource for Enhanced Decisions in Care and Treatment (PREDICT) [6] using Illumina’s VeraCode ADME core panel (Illumina, Inc. San Diego, CA, USA) [18] and in five U.S. medical centers [7]. However, there are several drawbacks to this technology in the context of pharmacogenomics testing. Firstly, novel variants of potential clinical relevance are not taken into consideration while using pharmacogenomic arrays. Several studies have shown that rare variants comprise 30–40% of the variation in pharmacogenes [19,20]. Secondly, due to the difference in test designs across different platforms, it becomes difficult to compare results and often leads to inconsistent haplotype calling for the same alleles [21]. Another problem that has been reported with the use of PGx arrays is in the identification of copy number variations (CNV) [21,22].

Next-generation sequencing (NGS) approaches are gaining more popularity in PGx, involving either pharmacogene-targeted/whole exome sequencing (WES) [23] or whole genome sequencing (WGS) [24]. The advantage of WGS is that in a single assay, it can detect not only disease-causing but also pharmacogenetically relevant variants. Patrinos et al., in their study analyzing 482 whole genome sequences, demonstrated the pre-eminence of WGS over other genetic screening methods to accurately determine an individual’s pharmacogenomic profile in a comprehensive manner [25]. A distinctive benefit of NGS technology is the ability to detect novel and rare variants in the genome, that might be missed in an array [26]. Furthermore, it yields better quantitative results with somatic variation as compared with Sanger sequencing technology and result in a higher throughput scale [27]. Whole exome sequencing, though it may appear as a viable choice compared to whole genome sequencing in terms of cost, fails to capture the regulatory and untranslated regions in the genome where many PGx variants reside. To further complicate choices, the efficiency of commercial target kits varies considerably, leaving a significant proportion of variants undetected [28,29]. Several studies, including that by Reisberg et al., have concluded that whole exome sequencing is not suitable for pharmacogenomic predictions [30]. One major challenge in the implementation of pharmacogenomics is the retrieval of genotypic marker information in star allele diplotype format. Some of the pharmacogenomic translation tools that are currently in use include Astrolabe [31], Aldy [32], Stargazer [33] and PharmCAT [34]. Except for PharmCAT, the other three tools work only in Linux and Mac Operating Systems and their output includes diplotypes, phenotypes, suballeles and novel Single-nucleotide variants (SNV) [35]. While Astrolabe allows both GRCh37 and GRCh38 input file formats, Aldy and Stargazer can only accept files in GRCh37 format, whereas PharmCAT allows Variant Call Format (VCF) only in GRCh38 format. Also, among the four, only PharmCAT provides drug guideline recommendations [34].

The next challenge is the translation of genetic test results into clinical action. PharmGKB has published PGx-based drug dosing guidelines by several consortia, including the Clinical Pharmacogenetics Implementation Consortium (CPIC) [36], the Dutch Pharmacogenetics Working Group (DPWG) [37,38], the Canadian Pharmacogenomics Network for Drug Safety (CPNDS) and other professional societies that provide therapeutic recommendations for well-known pharmacogene–drug pairs. A comparison study between CPIC and DPWG guidelines reported substantial similarities and few observed differences that could lead to the use of different methodologies for drug dosing [22,39].

Finally, the success of PGx implementation relies heavily on its acceptance among patients and clinical healthcare professionals. The major stumbling block to its widespread implementation among general physicians and clinical geneticists appears to be the lack of knowledge of genetics and an unfamiliarity with PGx data and tools. CDS delivered through electronic health records (EHRs) has proved indispensable in facilitating gene-based drug prescription for patient care [26,40].

Considering the current state of clinical pharmacogenomics together with the availability of pharmacogenomic resources, we have created a web-based tool that facilitates the easy transition of a person’s whole genome variant data into clinical recommendations. Through this, we aim to enable clinicians and geneticists to more broadly implement pharmacogenomics in patient care. The initial version of this software covers nine well-annotated pharmacogenes that cover the activity of 37 drugs.

## 2. Materials and Methods

### 2.1. Pharmacogenes

Gene-specific information tables provided jointly by PharmGKB and CPIC [36] were used to finalize the pharmacogenes used in this tool. We restricted the number of genes to only those for which a diplotype to phenotype information table was available and were common to the 28 genes mentioned in the study by Lee et al. describing the utility of Stargazer on whole genome sequences [33].

### 2.2. Calling Star Alleles

Genetic markers in pharmacogenomics are indicated using star-allele nomenclature—numbers and letters and separated from the gene name by a star. Several bioinformatics software tools that aid in the conversion of a genome variants to star-allele nomenclature are available including Astrolabe [31], PharmCAT [34,41] and Stargazer [42]. We examined concordance in calling star-allele nomenclature, by way of testing these tools on 20 in-house whole-genomes with coverage greater than 30X in order to select the tool most suitable for our purpose.

### 2.3. Diplotype–Phenotype Mapping

We used the diplotype-phenotype table, from the gene-specific information tables, to map the sample diplotype assigned in the previous step to its phenotype. Allele functionality data was also obtained from the diplotype-phenotype table. Medication/drug name information for the corresponding gene was obtained from the Clinical Pharmacogenetics Implementation Consortium (CPIC) of the National Institutes of Health’s Pharmacogenomics Research Network (http://www.pgrn.org) (accessed on 7 September 2020). Drug dosage information was retrieved from the Pharmacogenomics Knowledge Base (PharmGKB, http://www.pharmgkb.org) (accessed on 15 December 2020).

### 2.4. Implementation

PharmaKU was implemented in Python3 and uses a Django web framework. It was deployed in Apache and mod_wsgi. PharmaKU is supported by all major browsers.

## 3. Results

### 3.1. List of Genes Included in the Tool

We chose 9 out of the 18 pharmacogenes listed in the gene-specific information tables based on the standard annotations that are available for each gene (Table 1).

### 3.2. Choice of Pharmacogenomic Tool for Calling Diplotypes

We compared three pharmacogenomic translation tools for calling diplotypes across the nine pharmacogenes in Table 1 in 20 WGS samples. We found that Stargazer called diplotypes in more genes; 92.2% of the cases compared with Astrolabe (33.3%) and PharmCAT (31.1%) (Table 2). We also observed better concordance in results between Stargazer and Astrolabe and Stargazer and PharmCAT than between Astrolabe and PharmCAT in any single sample. For these reasons, we decided to implement Stargazer version 1.2.2 in our software for calling star alleles from the nine pharmacogenes using WGS data. We have also assessed five samples independently using two different technologies: Illumina’s pharmacogenetic-targeted panel and whole genome sequence data. Scoring showed 100% accuracy between the two methods (data not shown).

### 3.3. Drugs and Dosing Information

Based on the analysis of the nine pharmacogenes in Table 1, we identified 49 drugs from PharmGKB PGx prescribing information for which CPIC dosing guidelines were available. Twelve of these drugs did not have any prescription recommendation and we included the remaining 37 drugs with their pharmacogenomics-based dosage recommendations in our pipeline (see Appendix A). More than one third of these drugs were categorized as antidepressants (38%), followed by alimentary tract and metabolism-related drugs (13%) and cancer drugs (11%) (Figure 1).

### 3.4. Using PharmaKU Software

Users can input the individual WGS VCF files through the web portal, which can be accessed securely from: http://ppgr.dasmaninstitute.org/. Access can be provided upon request.

It is assumed that all VCF inputs meets minimum quality requirements and have a coverage of at least 30X. Files should be single sample VCF files in hg19 or GRCh38 reference format. The diplotypes called and the authenticity of the final report largely depend on the credibility of the input file.

In the background, the software performs two tasks (Figure 2). The main task involves the following steps: inferring diplotypes for the nine pharmacogenes based on the input VCF file and then retrieving information corresponding to the called diplotype from PharmGKB and the CPIC. The process gathers phenotype and allele functionality information for the corresponding gene–diplotype pair from diplotype-phenotype tables obtained from PharmGKB. From the list of 37 drugs that are affected by the nine pharmacogenes, and for which CPIC drug dosage guidelines are available, the dosage recommendations are updated in the final report. These drugs have the FDA label “Actionable” or “Informative”, and are drugs for which PharmGKB has made available pre- and post-test alerts and flowcharts.

The final report consists of two sections. The first section gives a summary of the genes with identified diplotype calls; allele functionality corresponding to the star alleles (unknown/uncertain/normal/no/increased/decreased function); phenotype status corresponding to allele functionality (indeterminate/poor/normal/intermediate/rapid/ultrarapid metabolizer) and clinical recommendations suggesting usual dose (for normal metabolizer) or adjust dose (for other phenotypes). There is an exception in the nomenclature of phenotype status in the SLCO1B1 gene according to the diplotype-phenotype file, where instead of the above, the conventions used are indeterminate/possibly decreased/decreased/normal/possibly increased/increased/possibly poor/poor function. The second section provides a detailed interpretation of the findings (consult note). For each gene listed in the first section, this will detail the consequence of the genotype on the allele functionality and phenotype. Wherever possible, dosage suggestions and changes are also recommended (see a sample report in Appendix A).

### 3.5. Report from 20 WGS Samples

We compiled the reports generated from 20 WGS VCF files to determine the type of metabolizers and phenotype status for seven of the nine pharmacogenes found in these individuals (Figure 3). DPYD and UGT1A1 were left out, as there was no phenotype status corresponding to the called diplotypes. It was observed that, in most of the genes, the majority of the samples exhibited normal metabolizer activity. However, for CYP3A5, almost 80% of the samples were poor metabolizers. CYP3A5 has known variants that modulate the activity of the drug tacrolimus, an antirejection medication for liver transplantation [43]. One of the reported complications in interpreting CYP3A5 genotyping results is that most of the individuals involved in drug trials were of European descent, and were therefore more likely to have the CYP3A5*3/*3 genotype, which predicts a poor metabolizer status. Hence, unlike other CYP enzymes, CYP3A5 variant and tacrolimus prescription pose an exception, wherein a CYP3A5 expresser (normal or intermediate metabolizer) would require a higher recommended starting dose and a CYP3A5 non-expresser (poor metabolizer) would require the standard recommended starting dose [44].

### 3.6. Data Collaboration and Testing

With this initial version of the software, we aim to encourage data collaboration and testing within the pharmacogenomics community. Upon user consent, input data may be used for collaborative research work and for clinical validation of the software. Users may also contact authors to communicate errors or issues faced while using the software.

## 4. Discussion

Advances in NGS have revolutionized the field of pharmacogenomics by pinpointing genetic variants relevant to drug action and metabolism. It is rightly said that pharmacogenomics is a forerunner in bringing precision medicine to the clinic. However, identification of variants is only the first step toward better treatment. The availability of quality-controlled and patient-centered software to link the identified pharmacogenomic variants from an individual’s genome to the existing knowledge of drug dosing guidelines holds the key to widespread and successful implementation of pharmacogenomics in our healthcare system [45]. In a study that evaluated the impact of preemptive pharmacogenomic genotyping results, an institutional CDS system provided pharmacogenomic results using traffic light alerts. As a result, medications with high pharmacogenomic risk were changed and no high-risk drugs were prescribed during the entire study [46].

We have taken the following measures to minimize false-positive results using our software. First, an important step in the pharmacogenomic translation process, prone to erroneous results, is the assigning of diplotypes. We shortlisted three currently available public tools and benchmarked them using our in-house data. Based on our results (Table 2), we incorporated Stargazer into our software for diplotype calling. Second, the remaining processes in the software pipeline deal with the mapping of assigned diplotype to a gene’s allele functionality, phenotype and dosage recommendation. We have retrieved this information directly from PharmGKB and CPIC resources, without using any third-party software, thereby minimizing chances of data corruption. Third, in recommending prescriptions, we have maintained a standard, wherein only drugs having CPIC guidelines and with pre- and post-test alert flowcharts in PharmGKB were included in our software. This was done to minimize discrepancies in naming and dosage information across different guidelines.

As of February 2021, dosage recommendations for nine pharmacogenes and 37 drugs have been incorporated into our software. The number of drugs for which prescription information is available is limited by the information provided by the CPIC guidelines. We plan to update the software with more genes in a timely manner. We also plan to adopt the drug dosing guidelines published by DPWG and CPNDS and make these recommendations available through our software in future versions. To the best of our knowledge, there is no other web-based, publicly available pharmacogenomics software that allows for VCF inputs in both hg19 and GRCh38 format. Through this effort, we have tried to simplify the pharmacogenomic translation process to the advantage of physicians, that with a single click, they are provided with a comprehensive pharmacogenomic report of their patient, complete with prescribing recommendations.

We have observed that more than one third of the drugs for which PGx dosing recommendations are available belong to the class of antidepressants. Lack of pharmacogenomic data on other commonly used drug classes, such as alimentary tract and metabolism-related drugs (13%), cancer drugs (11%) and cardiovascular/lipid-modifying drugs (3%), will shed more light on the necessity for more studies in this field.

Limitation: Although the cost of WGS continues to decline, it remains prohibitively expensive for widespread clinical use. However, its use is justified by the fact that a one-time genomic test to determine a person’s pharmacogenomic profile would inform clinicians about dosing and effectiveness for a multitude of drugs. Inclusion of this information into the EHR would be invaluable to patients throughout their lifetime.

## Figures and Tables

**Figure 1 jpm-11-00210-f001:**
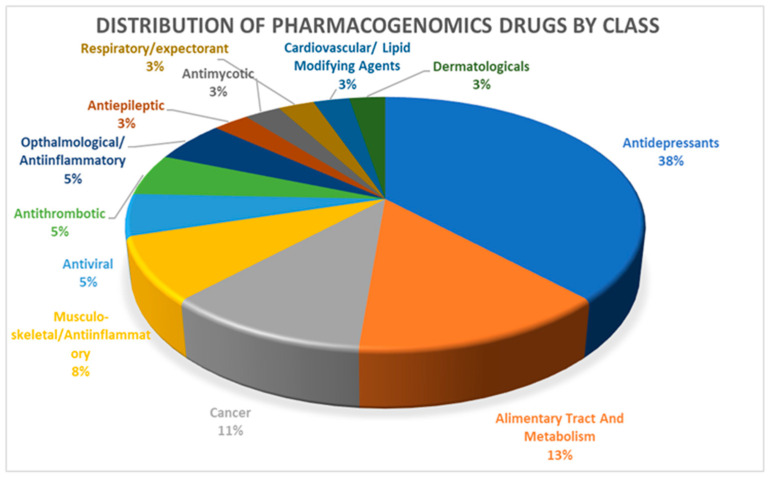
Distribution of 37 pharmacogenomic drugs by class.

**Figure 2 jpm-11-00210-f002:**
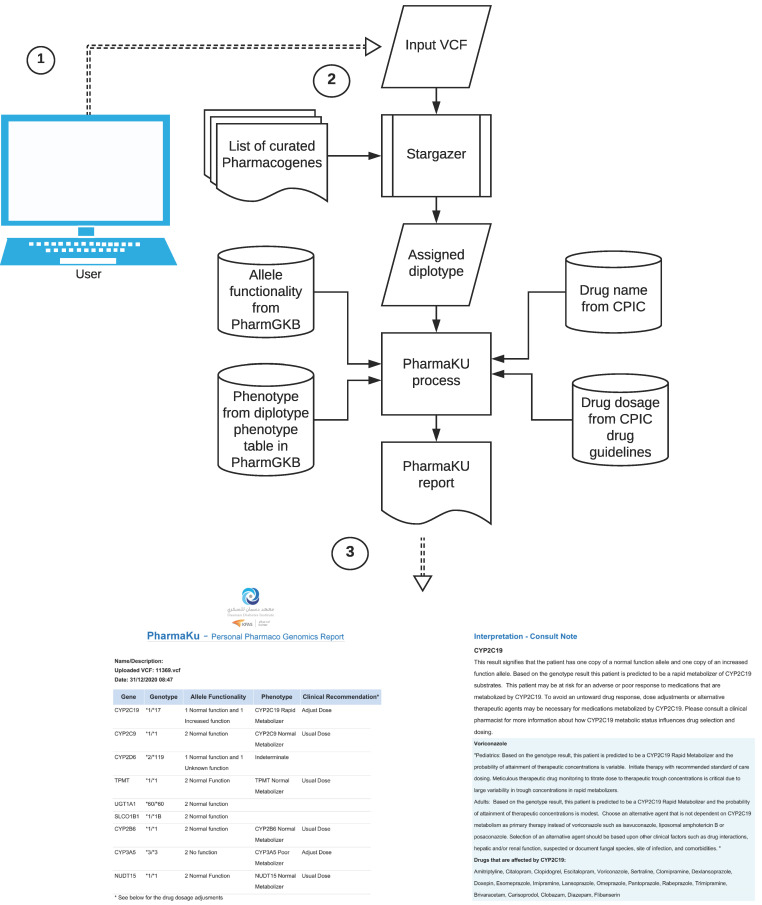
Working of the PharmaKU software: ① Single sample WGS VCF file provided as user input through the web interface; ② The software uses Stargazer to detect diplotypes in the nine pharmacogenes and assigns the corresponding allele functionality, phenotype, drug names and recommended dosage information; ③ Personalized pharmacogenomics report generated that is downloadable in PDF format.

**Figure 3 jpm-11-00210-f003:**
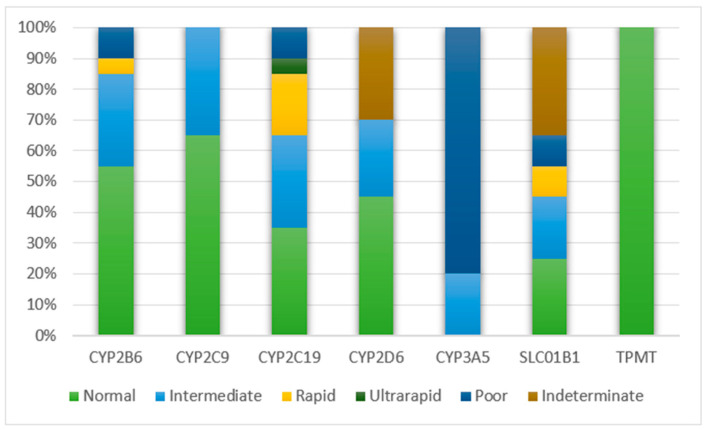
Metabolizer type for major pharmacogenes from 20 individual reports.

**Table 1 jpm-11-00210-t001:** List of nine pharmacogenes used in PharmaKU along with associated drugs.

Gene	Drug	PGx on FDA Label	CPIC Publications (PMID)
CYP2B6	efavirenz	Actionable PGx	31006110
CYP2C19	amitriptyline		23486447; 27997040
	citalopram	Actionable PGx	25974703
	clopidogrel	Actionable PGx	21716271; 23698643
	escitalopram	Actionable PGx	25974703
	lansoprazole	Informative PGx	32770672
	omeprazole	Actionable PGx	32770672
	pantoprazole	Actionable PGx	32770672
	voriconazole	Actionable PGx	27981572
	clomipramine		23486447; 27997040
	dexlansoprazole	Actionable PGx	32770672
	doxepin	Actionable PGx	23486447; 27997040
	imipramine		23486447; 27997040
	sertraline		25974703
	trimipramine		23486447; 27997040
	esomeprazole	Actionable PGx	32770672
	rabeprazole	Actionable PGx	32770672
CYP2C9	celecoxib	Actionable PGx	32189324
	flurbiprofen	Actionable PGx	32189324
	fosphenytoin		25099164; 32779747
	ibuprofen		32189324
	lornoxicam		32189324
	meloxicam	Actionable PGx	32189324
	phenytoin	Actionable PGx	25099164; 32779747
	piroxicam	Actionable PGx	32189324
	tenoxicam		32189324
	warfarin	Actionable PGx	21900891; 28198005
	aceclofenac		32189324
	aspirin		32189324
	diclofenac		32189324
	indomethacin		32189324
	lumiracoxib		32189324
	nabumetone		32189324
	naproxen		32189324
CYP2D6	amitriptyline	Actionable PGx	23486447; 27997040
	atomoxetine	Actionable PGx	30801677
	codeine	Actionable PGx	22205192; 24458010
	nortriptyline	Actionable PGx	23486447; 27997040
	ondansetron	Informative PGx	28002639
	paroxetine	Informative PGx	25974703
	tamoxifen	Actionable PGx	29385237
	tropisetron		28002639
	clomipramine	Actionable PGx	23486447; 27997040
	desipramine	Actionable PGx	23486447; 27997040
	doxepin	Actionable PGx	23486447; 27997040
	fluvoxamine	Actionable PGx	25974703
	imipramine	Actionable PGx	23486447; 27997040
	trimipramine	Actionable PGx	23486447; 27997040
CYP3A5	tacrolimus		25801146
DPYD	capecitabine	Actionable PGx	23988873; 29152729
	fluorouracil	Actionable PGx	23988873; 29152729
	tegafur		23988873; 29152729
SLCO1B1	simvastatin		22617227; 24918167
TPMT	azathioprine	Testing recommended	21270794; 23422873; 30447069
	mercaptopurine	Testing recommended	21270794; 23422873; 30447069
	thioguanine	Testing recommended	21270794; 23422873; 30447069
UGT1A1	atazanavir		26417955

**Table 2 jpm-11-00210-t002:** Comparison of diplotype detected in the nine pharmacogenes using Astrolabe, PharmCAT and Stargazer in 20 whole genome sequencing (WGS) samples.

#	Sample_ID	Tool	Gene								
			CYP2B6	CYP2C9	CYP2C19	CYP2D6	CYP3A5	DPYD	SLC01B1	TPMT	UGT1A1
1	1	Astrolabe		*1/*2	*1/*1	*2/*4					
		PharmCAT		*1/*2, *1/*35					*5/*20, *5/*21		*36, *60, *60
		Stargazer	*1/*2	*1/*2	*1/*1	*2/*4	*3/*3	*S12/*S12	*1/*5	*1/*1	
2	2	Astrolabe		*1/*1	*1/*17	*2/*41					
		PharmCAT			*1/*4B, *1/*17				*1A/*18		*60/*60
		Stargazer	*1/*1	*1/*1	*1/*17	*2/*119	*3/*3	*6/*S12	*1/*1B	*1/*1	*60/*60
3	3	Astrolabe		*1/*1	*2/*2	*41/*86					
		PharmCAT	*2/*2						*19/*20, *19/*21		*36, *60
		Stargazer	*1/*6	*1/*1	*2/*2	*86/*119	*3/*3	*1/*9A	*1/*1B	*1/*1	
4	4	Astrolabe		*1/*1	*1/*17	*1/*41					
		PharmCAT			*1/*4B, *1/*17						*36, *60
		Stargazer	*1/*1	*1/*1	*1/*17	*1/*119	*1/*3	*S3/*5	*1/*14	*1/*1	
5	5	Astrolabe		*1/*1	*1/*2	*10/*4					
		PharmCAT			*1/*2				*1A/*20, *1A/*21		
		Stargazer	*1/*22	*1/*1	*1/*2	*4/*10	*1/*3	*S3/*S12	*1/*1B	*1/*1	*79/*79
6	6	Astrolabe		*1/*1	*1/*2	*1/*86					
		PharmCAT			*1/*2				*1A/*18		*60/*60
		Stargazer	*5/*6	*1/*1	*1/*2	*1/*1	*3/*3	*S3/*S12	*1/*1B	*1/*1	
7	7	Astrolabe		*2/*17	*1/*1	*1/*2					
		PharmCAT			*2/*4B, *2/*17				Multiple		
		Stargazer	*1/*1	*1/*1	*2/*17	*1/*2	*3/*3	*1/*S12	*1/*S464F	*1/*1	*79/*79
8	8	Astrolabe		*1/*1	*1/*1	*1/*10					
		PharmCAT							*18/*18, *18/*19, *19/*19		*60
		Stargazer	*6/*6	*1/*1	*1/*1	*1/*10	*1/*3	*S12/*S38	*1/*1	*1/*1	*60/*79
9	9	Astrolabe		*1/*2	*1/*1	*1/*4					
		PharmCAT			*1/*2				*1A/*18, *1A/*19		*36, *60
		Stargazer	*1/*1	*1/*1	*1/*2	*1/*4	*3/*3	*S12/*S12	*1/*1	*1/*1	
10	10	Astrolabe		*2/*2	*1/*1	*1/*4					
		PharmCAT			*2/*2				*20/*20, *20/*21, *21/*21		*60/*60
		Stargazer	*6/*6	*1/*1	*2/*2	*1/*4	*1/*3	*1/*S12	*1B/*1B	*1/*1	
11	11	Astrolabe		*2/*17	*1/*1	*1/*2					
		PharmCAT			*2/*4B, *2/*17				rs4149056T/rs4149056C		*36, *60
		Stargazer	*1/*5	*1/*1	*2/*17	*1/*2	*3/*3	*9A/*S12	*1/*17	*1/*1	
12	12	Astrolabe		*1/*1	*1/*2	*2/*4					
		PharmCAT			*1/*2, *1/*35				*5/*20, *5/*21		*36, *60
		Stargazer	*1/*1	*1/*2	*1/*1	*2/*4	*3/*3	*6/*S12	*1/*15	*1/*1	
13	13	Astrolabe		*1/*1	*1/*2	*1/*1					
		PharmCAT		*1/*2, *1/*35							
		Stargazer	*1/*6	*1/*2	*1/*1	*1/*122	*3/*3	*5/*9A	*1/*14	*1/*1	*1/*79
14	14	Astrolabe		*1/*1	*1/*3	*1/*1					
		PharmCAT		*1/*3, *1/*18					rs4149056C/rs4149056C		*36, *60
		Stargazer	*2/*6	*1/*3	*1/*1	*1/*1	*3/*3	*5/*S12	*15/*15	*1/*1	
15	15	Astrolabe		*1/*1	*1/*1	*1/*41					
		PharmCAT							*1A/*18, *1A/*19		*36, *60, *60
		Stargazer	*1/*5	*1/*1	*1/*1	*1/*119	*3/*3	*S12/*S12	*1/*1	*1/*1	
16	16	Astrolabe		*1/*2	*1/*2	*1/*41					
		PharmCAT		*1/*2, *1/*35	*1/*2				rs4149056C/rs4149056C		*36, *60
		Stargazer	*1/*9	*1/*2	*1/*2	*1/*119	*3/*3	*9A/*9A	*15/*15	*1/*1	
17	17	Astrolabe		*17/*17	*1/*1	*1/*2					
		PharmCAT			*4B/*4B, *4B/*17, *17/*17				rs4149056T/rs4149056C		*60/*60
		Stargazer	*1/*6	*1/*1	*17/*17	*1/*2	*3/*3	*9A/*S12	*1/*17	*1/*1	
18	18	Astrolabe		*1/*17	*1/*1	*1/*1					
		PharmCAT			*1/*4B, *1/*17				*18/*18, *18/*19, *19/*19		*36, *60
		Stargazer	*1/*1	*1/*1	*1/*17	*1/*1	*3/*3	*9A/*S12	*1/*1	*1/*1	
19	19	Astrolabe		*1/*1	*1/*2	*1/*2					
		PharmCAT		*1/*2, *1/*35					*1A/*18		*36, *60
		Stargazer	*1/*1	*1/*2	*1/*1	*1/*2	*3/*3	*9A/*9A	*1/*1B	*1/*1	
20	20	Astrolabe		*1/*17	*1/*2	*2/*2					
		PharmCAT		*1/*2, *1/*35	*1/*4B, *1/*17				*1A/*18		
		Stargazer	*1/*1	*1/*2	*1/*17	*2/*2	*3/*3	*9A/*9A	*1/*1B	*1/*1	*1/*1

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
