# Peer review of "PharmaKU: A Web-Based Tool Aimed at Improving Outreach and Clinical Utility of Pharmacogenomics"

_jpm, 2021, doi:10.3390/jpm11030210_

Round 1
Reviewer 1 Report
Sumi et al., summarized key findings in this manuscript entitled, “PharmaKU: a web-based tool aimed at improving outreach and clinical utility of Pharmacogenomics” holds scientific merits in the personalized medicine field. However, after careful review, I think authors may consider following comments….
Major comments:
- Authors need to show/ discuss clinical validation of the report by which clinicians will learn and implement without false-positive/negative recommendations.
- Why authors only included the genes diplotype-phenotype table available instead of collecting POC validation and clinical implemented examples form other hospitals/ country? Does, authors have plan for clinical validation?
- What is the main advantage of this PharmaKU form others? If YES, then please discuss and show POC-validation in the discussion for the readers.
- How PharmaKU will work on dosage suggestions? Author needs to describe in details for this output, based on what and rare confounding factors, disease-drug interactions etc.
- Why authors are only considering WGS, instead of exon/intron junction sequencing for making it less costly? Even to date, there are genotype-phenotype data available for varieties of SNPs. From this, target gene (specific region in the chromosomes) is possible. Author may consider discussing on this issue where pre-emptive genotyping is not covered by insurance.
- Author can include output template as supplementary or into main body table for better understanding the data and format
Minor comments:
- Line 34-35, cite reference
- Line 55-57, cite reference
- Please cite other clinical implementation program (i.e Netherlands group) in the discussion. Otherwise discussion and future direction is very limited at current version
Author Response
Major comments:
Point 1: Authors need to show/ discuss clinical validation of the report by which clinicians will learn and implement without false-positive/negative recommendations.
Response 1: We have taken the following measures to minimize false-positive results using our software:
- An important step in the pharmacogenomic translation process, which is prone to erroneous results is the assigning of diplotypes. We shortlisted three currently available public tools and benchmarked them using our in-house data. Based on our results (Table 2), we incorporated Stargazer into our software for diplotype calling.
- The remaining processes in the software pipeline deals with mapping of assigned diplotype to gene’s allele functionality, phenotype and dosage recommendation. We have retrieved this information directly from PharmGKB and CPIC resources, without using any third-party software, thereby minimizing chances of data corruption.
- For applying prescription recommendations, we have maintained a standard, wherein only drugs having CPIC guidelines and with pre- and post-test alerts flowcharts in PharmGKB were included in our software. This was done to minimize discrepancies in naming and dosage information across different guidelines.
The following sentences have been added to the Discussion section in the manuscript (lines 263-275):
We have taken the following measures to minimize false-positive results using our software: i. An important step in the pharmacogenomic translation process, which is prone to erroneous results is the assigning of diplotypes. We shortlisted three currently available public tools and benchmarked them using our in-house data. Based on our results (Table 2), we incorporated Stargazer into our software for diplotype calling. ii. The remaining processes in the software pipeline deals with mapping of assigned diplotype to gene’s allele functionality, phenotype and dosage recommendation. We have retrieved this information directly from PharmGKB and CPIC resources, without using any third-party software, thereby minimizing chances of data corruption. iii. For applying prescription recommendations, we have maintained a standard, wherein only drugs having CPIC guidelines and with pre- and post-test alerts flowcharts in PharmGKB were included in our software. This was done to minimize discrepancies in naming and dosage information across different guidelines.
Point 2: Why authors only included the genes diplotype-phenotype table available instead of collecting POC validation and clinical implemented examples form other hospitals/ country? Does, authors have plan for clinical validation?
Response 2: With this initial version of the software, we aim to encourage data collaboration and testing within the pharmacogenomics community. Upon user consent, input data may be used for collaborative research work and for clinical validation of the software. Users may contact authors to communicate errors or issues faced while using the software.
The following section has been added to the Results section in the manuscript (lines 245-248):
3.6 Data collaboration and testing
With this initial version of the software, we aim to encourage data collaboration and testing within the pharmacogenomics community. Upon user consent, input data may be used for collaborative research work and for clinical validation of the software. Users may contact authors to communicate errors or issues faced while using the software.
Point 3: What is the main advantage of this PharmaKU form others? If YES, then please discuss and show POC-validation in the discussion for the readers.
Response 3: The main advantage that PharmaKU offers compared to other pharmacogenomic tools is that it is an end-to-end web-based tool that could be used by physicians as a means to implement precision medicine in clinics, allowing them to consider the genetic profile of the patients while prescribing medications and facilitating clinical decision making. To the best of our knowledge, there is no other web-based, publicly available pharmacogenomics software that allows VCF input in both hg19 and GRCh38 format. Through this effort we have tried to simplify the pharmacogenomic translation process to the advantage of treating physicians, that with a single click, they are provided with a comprehensive pharmacogenomic report of their patient, complete with prescribing recommendations.
The following sentences have been added to the Discussion section in the manuscript (lines 281-286):
To the best of our knowledge, there is no other web-based, publicly available pharmacogenomics software that allows VCF input in both hg19 and GRCh38 format. Through this effort we have tried to simplify the pharmacogenomic translation process to the advantage of treating physicians, that with a single click, they are provided with a comprehensive pharmacogenomic report of their patient, complete with prescribing recommendations.
Point 4: How PharmaKU will work on dosage suggestions? Author needs to describe in details for this output, based on what and rare confounding factors, disease-drug interactions etc.
Response 4: PharmaKU extracts prescribing recommendations for drugs based on genetic profile of the individual from PharmGKB, which annotates drug dosing guidelines by three established groups- CPIC, DPWG and CPNDS. In this version of PharmaKU, we have tried to minimize the confounding factors by adhering to CPIC guidelines alone. Moreover, it has been validated by a study by Bank et al. that observed substantial similarities and few observed differences between CPIC and DPWG dosing guidelines. Furthermore, in the Interpretation section of the report, for each gene, we have retrieved the Interpretation text from the Diplotype-Phenotype table, based on the individual’s phenotype (Metabolizer type). For each gene, we have also listed the associated drugs that have the FDA label “Actionable” or “Informative”, and for which PharmGKB has made available Pre- and Post-Test alerts and flowcharts.
The following sentences have been added to the Results section in the manuscript (lines 202-204):
These drugs have the FDA label “Actionable” or “Informative”, and for which PharmGKB has made available Pre- and Post-Test alerts and flowcharts.
Point 5: Why authors are only considering WGS, instead of exon/intron junction sequencing for making it less costly? Even to date, there are genotype-phenotype data available for varieties of SNPs. From this, target gene (specific region in the chromosomes) is possible. Author may consider discussing on this issue where pre-emptive genotyping is not covered by insurance.
Response 5: We have discussed this briefly in the Introduction, (Page 2, Line 71-77) on the drawbacks of using microarrays and whole exome sequencing (WES) (Page 2 Line 88-94) in pharmacogenomics. As pharmacogenomic profiling is based on star-based genotype calling method, accurate detection of star alleles is made possible by NGS methods. Several studies have shown that rare variants comprise 30-40% of the variation in pharmacogenes[1, 2]. This will go undetected with targeted sequencing methods.
The following sentence have been added to the Introduction section in the manuscript (lines 73-75):
Several studies have shown that rare variants comprise 30-40% of the variation in pharmacogenes.
And in References:
- Ingelman-Sundberg, M., et al., Integrating rare genetic variants into pharmacogenetic drug response predictions. Hum Genomics, 2018. 12(1): p. 26.
- Pandi, M.T., et al., Exome-Wide Analysis of the DiscovEHR Cohort Reveals Novel Candidate Pharmacogenomic Variants for Clinical Pharmacogenomics. Genes (Basel), 2020. 11(5).
Point 6: Author can include output template as supplementary or into main body table for better understanding the data and format
Response 6: We have now included a sample report in the supplementary data.
The following sentence have been added to the Results section in the manuscript (line 217): (see a sample report in supplementary data: SampleReport.pdf)
Minor comments:
Point 7: Line 34-35, cite reference
Response 7: Line 35: Inserted citation [2] in References: Primorac, D., et al., Pharmacogenomics at the center of precision medicine: challenges and perspective in an era of Big Data. Pharmacogenomics, 2020. 21(2): p. 141-156.
Point 8: Line 55-57, cite reference
Response 8: Line 57: Inserted citation [14] in References: Lang, T., et al., Extensive genetic polymorphism in the human CYP2B6 gene with impact on expression and function in human liver. Pharmacogenetics, 2001. 11(5): p. 399-415.
Point 9: Please cite other clinical implementation program (i.e Netherlands group) in the discussion. Otherwise discussion and future direction is very limited at current version
Response 9: Added this in Discussion Lines 257-262: In a study that evaluated the impact of preemptive pharmacogenomic genotyping results, an institutional clinical decision support (CDS) system provided pharmacogenomic results using traffic light alerts. As a result, medications with high pharmacogenomic risk were changed as part of intervention and no high-risk drugs were prescribed during the entire study[47].
[47] in References: O'Donnell, P.H., et al., Pharmacogenomics-Based Point-of-Care Clinical Decision Support Significantly Alters Drug Prescribing. Clin Pharmacol Ther, 2017. 102(5): p. 859-869.
References
- Ingelman-Sundberg, M., et al., Integrating rare genetic variants into pharmacogenetic drug response predictions. Hum Genomics, 2018. 12(1): p. 26.
- Pandi, M.T., et al., Exome-Wide Analysis of the DiscovEHR Cohort Reveals Novel Candidate Pharmacogenomic Variants for Clinical Pharmacogenomics. Genes (Basel), 2020. 11(5).
Reviewer 2 Report
The authors introduced PharmaKU, a user-friendly web-based pharmacogenomics tool, in this paper.The tool allows us to select both hg19/GRCh37 and hg38/GRCh38, and it is also attractive to be able to report the results in PDF format.
1) P4 151-152 3.1 List of genes included in the tool.
The authors stated that "We chose 9 out of the 18 pharmacogenes″.
Please describe more detail what criteria the authors to select the 9 pharmacogenes (CYP2B6, CYP2C9, CYP2C19, CYP2D6, CYP3A5, DYPD, SLCO1B1, TPMP and UGT1A1).
There are other clinically important genes polymorphisms (Ex; such as HLA, VKORC1 and NUDT15 etc). Why are these not covered?
2) P4 154 Table 1
Table 1 lists a total of 52 drugs. On the other hand, Drugs and dosing information in 3.3 (P10), it is stated that "Based on the nine pharmacogenes, we identified 49 drugs from PharmGKB PGx Prescribing Info for which CPIC dosing guidelines were available." What is the difference from the drugs listed in Table 1?
3) P10 177-178 3.3 Drugs and dosing information
There is the following description, but the supplementary data created in Excel cannot be confirmed.
“see the list of drugs in supplementary data: guidelinesByDrugs.tsv.xlsx”
Is supplementary data presented?
I can't find it from this peer review system.
4) Figure 2 ③
It is better to show how the actual report will be presented in a way that the interested reader can understand it.
The generated personalized pharmacogenomics report (Figure 2 ③) is small size and readers for this journal might be difficult to understand.
5) Other
The description notation of GrCh and GRCh is mixed. Please unify them into GRCh.
Author Response
Point 1: P4 151-152 3.1 List of genes included in the tool.
The authors stated that "We chose 9 out of the 18 pharmacogenes″.
Please describe more detail what criteria the authors to select the 9 pharmacogenes (CYP2B6, CYP2C9, CYP2C19, CYP2D6, CYP3A5, DYPD, SLCO1B1, TPMP and UGT1A1).
There are other clinically important genes polymorphisms (Ex; such as HLA, VKORC1 and NUDT15 etc). Why are these not covered?
Response 1: PharmGKB and CPIC have jointly prepared and maintained Gene-specific information tables for 18 pharmacogenes. For each gene, they have provided the following resources as available: Allele Definition table, Allele Functionality table, Frequency table, Diplotype-Phenotype table and Gene Resource mappings. As we mentioned in P3 127-128, we chose only the genes for which Diplotype-Phenotype table is available and common with the 28 genes mentioned in the study by Lee et al. describing the utility of Stargazer on Whole genome sequences[1], since we chose Stargazer for calling star alleles. PharmGKB do not provide Diplotype-Phenotype information for VKORC1 and HLA genes. NUDT15 is not listed among the 28 genes. Hence, though important, these genes are not included in the list of genes for which PharmaKU annotates whole genome variants in the initial version of the software.
The following sentences have been added in the Results section in the manuscript (lines 130-131):
and common with the 28 genes mentioned in the study by Lee et al. describing the utility of Stargazer on Whole genome sequences[33]
Point 2: P4 154 Table 1
Table 1 lists a total of 52 drugs. On the other hand, Drugs and dosing information in 3.3 (P10), it is stated that "Based on the nine pharmacogenes, we identified 49 drugs from PharmGKB PGx Prescribing Info for which CPIC dosing guidelines were available." What is the difference from the drugs listed in Table 1?
Response 2: Table 1 lists all the drugs for which CPIC guidelines is available for the selected nine genes. In the PharmaKU report, we have included only those drugs that belongs to FDA approved “Actionable” or “Informative” PGx levels. 3 drugs in Table 1 (azathioprine, mercaptopurine and thioguanine) are labelled under the category “Testing recommended”, hence they were not used in the software for generating prescription information.
The following sentences have been added to the Results section in the manuscript (lines 202-204):
These drugs have the FDA label “Actionable” or “Informative”, and for which PharmGKB has made available Pre- and Post-Test alerts and flowcharts.
Point 3: P10 177-178 3.3 Drugs and dosing information
There is the following description, but the supplementary data created in Excel cannot be confirmed.
“see the list of drugs in supplementary data: guidelinesByDrugs.tsv.xlsx”
Is supplementary data presented?
I can't find it from this peer review system.
Response 3: Now added supplementary file with list of 37 drugs that is used in PharmaKU.
The following change has been made in the Results section in the manuscript (line 181):
Changed from “list of drugs in supplementary data: guidelinesByDrugs.tsv.xlsx” to
”list of drugs in supplementary data: DrugsList.xlsx”
Point 4: Figure 2 ③
It is better to show how the actual report will be presented in a way that the interested reader can understand it.
The generated personalized pharmacogenomics report (Figure 2 ③) is small size and readers for this journal might be difficult to understand.
Response 4: We have now included a sample report in the supplementary data.
The following sentence have been added to the Results section in the manuscript (line 217): (see a sample report in supplementary data: SampleReport.pdf)
Point 5: Other
The description notation of GrCh and GRCh is mixed. Please unify them into GRCh.
Response 5: Correction from GrCh to GRCh done in line numbers: 100,101,192.
References
- Lee, S.B., et al., Calling Star Alleles With Stargazer in 28 Pharmacogenes With Whole Genome Sequences. Clin Pharmacol Ther, 2019. 106(6): p. 1328-1337.